# Comparative Study of Air–Water and Air–Oil Frictional Pressure Drops in Horizontal Pipe Flow

**Enrique Guzmán** [1], **Valente Hernández Pérez** [2,†], **Fernando Aragón Rivera** [3,†], **Jaime Klapp** [4,*] and **Leonardo Sigalotti** [5]

1   Instituto de Ingeniería UNAM, Circuito Escolar S/N, Ciudad Universitaria, Ciudad de México 04510, Mexico; jguzmanv@iingen.unam.mx
2   Tecnológico Nacional de México, Pachuca de Soto 42080, Mexico; valente.hp@pachuca.tecnm.mx
3   Departamento de Energía, Universidad Autónoma Metropolitana, Unidad Azcapotzalco (UAM-A), Ciudad de México 02128, Mexico; micme2003@yahoo.com.mx
4   Departamento de Física, Instituto Nacional de Investigaciones Nucleares (ININ), Estado de México Ocoyoacac 52750, Mexico
5   Departamento de Ciencias Básicas, Universidad Autónoma Metropolitana, Unidad Azcapotzalco (UAM-A), Ciudad de México 02128, Mexico; leonardo.sigalotti@gmail.com
*   Correspondence: jaime.klapp@inin.gob.mx; Tel.: +52-55-3040-3724
†   These authors contributed equally to this work.

**Abstract:** Experimental data for frictional pressure drop using both air–water and air–oil mixtures are reported, compared and used to evaluate predictive methods. The data were gathered using the 2-inch (54.8 mm) flow loop of the multiphase flow facility at the National University of Singapore. Experiments were carried out over a wide range of flow conditions of superficial liquid and gas velocities that were varied from 0.05 to 1.5 m/s and 2 to 23 m/s, respectively. Pressure drops were measured using pressure transducers and a differential pressure (DP) cell. A hitherto unreported finding was achieved, as the pressure drop in air–oil flow can be lower than that in air–water flow for the higher range of flow conditions. Using flow visualization to explain this phenomenon, it was found that it is related to the higher liquid holdup that occurs in the case of air–oil around the annular flow transition and the resulting interfacial friction. This additional key finding can have applications in flow assurance to improve the efficiency of oil and gas transportation in pipelines. Models and correlations from the open literature were tested against the present data.

**Keywords:** pipe flow; air–water flow; air–oil flow; two-phase flow; pressure drop; friction factor

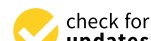



## 1. Introduction

Pressure drop prediction remains a challenge within the multiphase flow community. There is an extensive body of work in this area that spans over six decades. Comprehensive reviews have been provided recently by Xu et al. [1] and Mekisso [2], regarding correlations and experimental data of two-phase frictional pressure drop in isothermal horizontal flow. However, most experimental work has been carried out with air–water mixtures, in small facilities, and most papers have focused on the evaluation of models for pressure drop calculation, some of them from a third-party point of view (see, for instance, Dukler et al. [3], Idsinga et al. [4], Ferguson and Spedding [5], Tribbe and Müller-Steinhagen [6], Garcia et al. [7] and Xu et al. [1]). Several different correlations have been suggested for use by different authors. In particular, Dukler et al. [3] found that the Lockhart–Martinelli correlation, the oldest of the five correlations tested, shows the best agreement with a set of carefully culled experimental data on pressure drop, which seems reasonable since some theoretical bases for this correlation were derived later by Chisholm [8]. On the other hand, Xu et al. [1] concluded that the correlations developed by Müller-Steinhagen and Heck [9] and Sun and Mishima [10] fit the entire experimental data under varying conditions and are

recommended for use in two-phase pipe flow. However, their database was collected from experiments with channels with inner diameters from 0.0695 to 14 mm. They, as well as other authors, have also concluded that an improvement of presently available correlations and the development of new correlations are needed for more accurate predictions of two-phase frictional pressure drop.

It is well known that the total pressure drop in two-phase flow in a pipe consists of three contributions due to acceleration, friction and gravitational effects. For horizontal flows, the frictional pressure gradient usually forms the most significant contribution to the overall two-phase pressure drop. According to Ferguson and Spedding [5], this is the case when the total mass velocity $G_r < 2700$ kg m$^{-2}$ s$^{-1}$, as the accelerational pressure drop, due to density change, is negligible. For single-phase flows, the pressure drop is caused by shear stress at the wall, but for two-phase flows, the situation is by far more complex since it involves indefinitely deformable interfaces. Due to this complexity, there are diverse predictive approaches of the pressure drop. They can be classified into: direct empirical correlations, homogeneous model correlations, correlations based on a two-phase multiplier, flow-pattern-specific models and numerical or interfacial friction models.

In the homogeneous model, two-phase flow is treated as a pseudo single-phase flow with average properties. The slip ratio is equal to unity, as both liquid and gas phases move at the same velocity. The standard form of this model is based on the Darcy equation as follows:

$$\left(\frac{dP}{dL}\right)_f = \frac{2f\rho_m V_m^2}{d} \tag{1}$$

where $\rho_m$ is the mixture density, $V_m$ is the mixture velocity, $d$ is the pipe diameter and $f$ is the Fanning friction factor calculated from average mixture properties. For fully developed laminar flow (i.e., Re < 2000), the friction factor is given by 64/Re. For turbulent flow in smooth pipes, the Blasius equation can be used, which is given by $f = 0.079\text{Re}^{-0.25}$ in the range $3000 < \text{Re} < 10^5$. It should be noted that some references use the factor 0.316 instead of 0.079 in the Blasius equation because, in the former case, the pressure drop is divided by four. A model for the two-phase viscosity is needed to calculate the Reynolds number, as described by Awad and Muzychka [11]. The calculation of the friction factor is the main difference among the correlations of the homogeneous model group. Direct empirical correlations express the pressure drop directly as a function of known parameters, such as mass flux, mass quality or mixture density, by using curve fitting. Examples are the correlations developed by Beggs and Brill [12] and Müller-Steinhagen and Heck [9].

In the two-phase flow multiplier model, also called a separated model, the two-phase pressure drop is calculated from the single-phase pressure drop multiplied by a two-phase multiplier, based upon the concept that the pressure drop for the liquid phase must equal the pressure drop for the gas phase regardless of the flow pattern. Examples of these correlations are those developed by Lockhart and Martinelli [13] and Friedel [14]. They introduced the two-phase multiplier, $\Phi_F$, to define the ratio between the two-phase pressure gradient and the pressure gradient of either the gas or liquid phase.

$$\left(\frac{dp}{dL}\right)_{TP} = \Phi_F^2 \left(\frac{dp}{dL}\right)_L. \tag{2}$$

The ratio of the liquid phase to the gas phase pressure drop that would occur if either fluid were flowing alone in the pipe with the original flow rate of each phase are related to the parameter $\chi$, known as the Lockhart and Martinelli parameter, as

$$\left(\frac{dp}{dL}\right)_L = \chi^2 \left(\frac{dp}{dL}\right)_G. \tag{3}$$

The parameter $\chi$ is independent of the void fraction and is a measure of the degree to which the two-phase mixture is close to being a liquid, i.e., $\chi_u^2 \gg 1$, or to being a gas, i.e., $\chi_u^2 \ll 1$.

The subscript $u$ is sometimes used to mean that both phases are turbulent. Chisholm [8] provided a convenient relationship in the form of

$$\Phi_{TP}^2 = 1 + \frac{C}{\chi} + \frac{1}{\chi^2},\tag{4}$$

where $C$ is the Chisholm coefficient, which is defined for four gas–liquid flow regimes as $C = 20$ (turbulent–turbulent), $C = 12$ (turbulent–viscous), $C = 10$ (viscous–turbulent) and $C = 5$ (viscous–viscous).

A more complex approach is the mechanistic modeling. The first stage of this method is the prediction of the flow pattern. Therefore, the success of a comprehensive unified mechanistic approach depends upon the accurate prediction of the specific flow pattern. In addition, improvements might be required in the closure equations to obtain a fully accurate prediction. For instance, in the study by Holt et al. [15], a drift flux model is employed for bubbly flow, while slug flow is treated with a mechanistic description invoking a separate section around and between the Taylor bubbles. For annular flow, a phenomenological approach is used, utilizing descriptions for the rates of entrainment, deposition and film thickness. The numerical (or interfacial) friction approach is based on the two-fluid model. For pipe flow, the flow is assumed to be one-dimensional (1D). The mass and momentum conservation equations for both gas and liquid are solved numerically using an iterative procedure, as described by Issa [16], which yields cross-sectional average values of phase velocities, pressure and phase fraction along the pipe. Auxiliary relations representing the interfacial and shear stress forces are used, while different closure models are formulated for momentum transfer for each flow regime. A disadvantage of the phenomenological approach is that in practical situations, where, for example, high pressures and/or large pipe diameters occur, the present flow regime is often unknown [17]. In addition, numerical methods for two-phase flows suffer from a difficulty in obtaining data for their validation [18]. Moreover, the development of physics-based models can be thought of as being less mature. Thus, the correlation approach is still preferred for practical purposes. Some of the correlations have been developed for particular scenarios, such as for micro- and mini-channels [10], evaporating flows [19] and refrigeration [20].

Of particular importance is the role that fluid properties play in oil/gas production, which can vary from one well to another. There is also a wide range of applications that use different fluids, such as in chemical, power, refrigeration, air conditioning, oil/gas production and oil refining plants. The simple change in fluid from one mixture to another can result in a variation in several important properties, such as viscosity, density and surface tension. In fact, the viscosities measured for different heavy oils can vary by orders of magnitude. Since most of the predictive correlations of pressure drop are based on experimental data with air–water mixtures, as was reported by Mekisso [2], the extrapolation of these predictive methods to more industry relevant fluids might not work. Thus, more experimental data are required for validation, particularly for air–oil mixtures at high Reynolds numbers. For instance, pressure drop experimental data in horizontal air–water flow were reported by Hernández-Pérez [21]. In a more recent work, Hamad et al. [22] reported data for three different pipe diameters using air–water mixtures. Due to the short length of their pipes (1 m), the flow pattern was created in such a way that it was always homogeneous. Not surprisingly, they concluded that the homogeneous model for the prediction of the pressure drop was the best.

As a matter of fact, according to the two latest comprehensive reviews, only two references have reported air–oil pressure drops, which were all cited by Mekisso [2] and ignored by Xu et al. [19]. In particular, Dukler et al. [3] and Tribbe and Müller-Steinhagen [6] include gas–oil data from the industry field in their reports, but they do not give details. Additional separate data sets can be found from other sources. For example, Mattar and Gregory [23] investigated air–oil slug flow in a pipe slightly inclined upwards, including slug velocity, holdup and pressure gradient. On the other hand, Badie et al. [24] reported

data of two-phase gas–liquid pressure gradient measurements for extremely low liquid, while Gokcal et al. [25] investigated experimentally the effects of oil with high viscosity on the flow pattern, pressure gradient and liquid holdup. Oil viscosities from 0.181 to 0.587 Pa s were obtained by changing the oil temperature. Their experiments were performed on a flow loop with a test section of 50.8 mm ID and 18.9 m long horizontal pipe. Superficial liquid and gas velocities varied from 0.01 to 1.75 m s$^{-1}$ and from 0.1 to 20 m s$^{-1}$, respectively. They found that the pressure drop increases with the viscosity. A further experimental investigation on two-phase air/high-viscosity oil flow in a horizontal pipe was presented by Foletti et al. [26]. Similarly, Zhao et al. [27] found that the flow characteristics of high-viscosity oil and gas flow show several significant differences with those of low-viscosity liquid by experimenting with a pipe diameter of 26 mm and liquid superficial and gas velocities < 0.5 and 12 m s$^{-1}$, respectively. More recently, Farsetti et al. [28] investigated air–oil flow for different pipe inclinations, including the horizontal case. They obtained several parameters, including pressure drops for high-viscosity oil. However, the air superficial velocities were limited to only 1.5 m s$^{-1}$. Similarly, Abdulkadir et al. [29] reported pressure drop data using air–silicon oil within a limited range of superficial velocities.

Owing to the diversity of multiphase flow scenarios, other studies, such as those by Hernández-Pérez et al. [30] and Szalinski et al. [31], have compared air–water and air–oil flow, but only for vertical pipe flow, and no pressure drop data were reported, as they focused on Taylor bubble and phase distribution, respectively. Similarly, Foletti et al. [26] presented a comparison of flow pattern maps using air–water and air–viscous oil flow. A survey of the open literature shows that no comparison of the pressure drop between air–water and air–oil has been reported. Moreover, it can be observed that, despite their significance, gas–oil pressure drop data are scarce in the literature. A state-of-the-art multiphase flow loop facility is available at the National University of Singapore. Thus, the main objective of this work is to report unique experimental data of two-phase pressure drop from this facility for both air–water and oil–water mixtures. Since the general trend is to improve the accuracy of predictive models, a comparison is made of the present data with selected models in the literature. In addition, the physics behind the flow behavior under different fluids has been studied visually to understand and explain the results at a fundamental level.

## 2. Experimental Description

The overall operation of the flow loop facility has been previously described by Loh et al. [32]. The three-phase (air–water–oil) flow loop is equipped with a manifold into three different diameters of seamless, standard stainless-steel pipe (2, 4 and 6 inches), schedule 10. For the present work, the 2-inch (54.8 mm) line of the flow loop was selected. A schematic diagram of the facility is illustrated in the "supervisory control and data acquisition interface" shown in Figure 1.

A diagram of the test section is shown in Figure 2. It has a total length of 40 m and follows a rectangular shape. Measurements were taken after 14 m (i.e., pipe length over pipe diameter, $L/D \sim 280$) from the inlet. As described by Loh et al. [32], the two-phase flow mixtures are created by mixing air with water and air with oil. Air is supplied by two compressors connected in parallel to a receiver tank. Water and oil are supplied from a separator tank using a water and oil pump, respectively. Individual flow rates of air, water and oil are measured prior to the mixing section. The air is measured with a vortex-type flow meter, whereas both oil and water are measured with Coriolis-type flow meters. The mixing section consists of a concentric 2-inch pipe of air joining a 4-inch 90° bend with liquid in a mixing bend configuration, as shown in Figure 2. Check valves are used to prevent liquid going into the air line and vice versa. The air flow is calculated based on the test section pressure (2-inch P4 in Figure 1), which is in the loop itself at 14 m from the inlet. The calculation is carried out using an ideal gas equation. Since the facility is also being used as a reference to test the performance of third-party multiphase flow meters, the gas and liquid flow meters have been optimized for accuracy. The vortex gas flow meter

used in the present experiments has a measurement uncertainty of 1% of the indicated value, while the Coriolis liquid flow meters have a measurement uncertainty of ±0.3% of the indicated value. The flow is controlled automatically from the computer control system using a PID algorithm implemented in the NI LabVIEW software (version 2013).

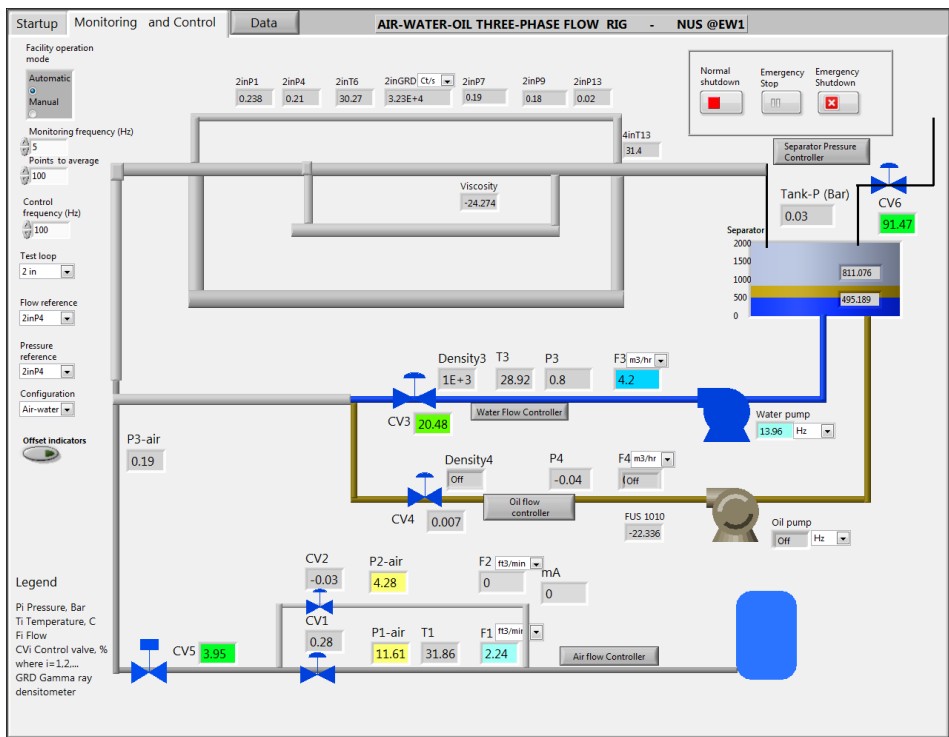

**Figure 1.** Monitoring, control and data acquisition system interface of the experimental setup.

After travelling across the flow loop, the mixture goes into a three-phase separator that also serves as a storage tank. After separation, the liquid is recycled, while air is released into the atmosphere through the control valve and a silencer. The separator tank volume is 16 m$^3$ and it holds 5 m$^3$ of water and 5 m$^3$ of oil. Efficient phase separation in the three-phase flow separator is confirmed by the density readings from the Coriolis flow meters.

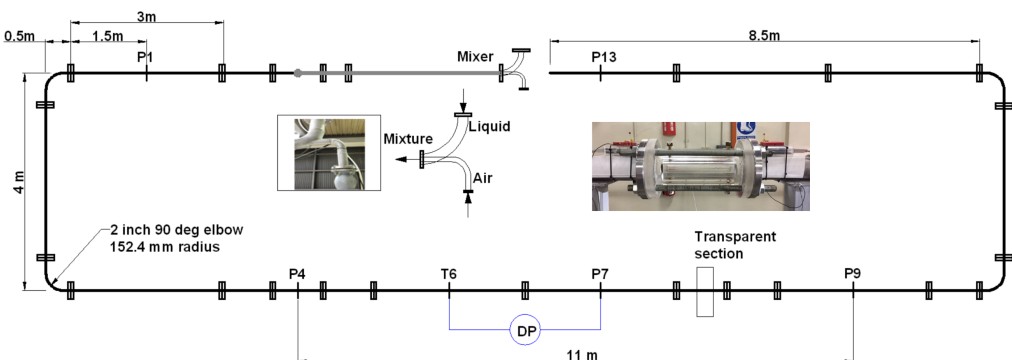

**Figure 2.** Top view of the physical configuration of the 2-inch test loop, mixing section and instrumentation.

## 2.1. Pressure Drop Measurement

Pressure and temperature sensors were placed at different axial locations along the test section, as shown in Figures 1 and 2. In addition, a DP cell was needed and used to read the lower pressure range (0–300 Pa m$^{-1}$). For the high range, the pressure drop was measured using two pressure sensors over a distance of 11 m of the straight pipe. All pressure taps were mounted flush in the tube wall. All sensors are factory calibrated and were tested

on site. Pressure sensors have a measurement uncertainty of 0.05% of the scale (16 Bar). Due to the high values of pressure drop for these flow conditions, in this large-scale facility, these kinds of pressure sensors are adequate. They may be advantageous as opposed to the case of the DP cell, where care must be taken to ensure that air is not present in the pressure line that would distort the measured value. However, this is hard to maintain under intermittent flow conditions. Other researchers, such as, for example, Moreno-Quiben and Thome [33], Ortiz-Vidal et al. [34] and Talley et al. [35], have also successfully obtained pressure drop measurements from absolute pressure sensors. The measurement locations were chosen to minimize the inlet and outlet effects while maintaining a total development length of at least 200 diameters. The measurements were used to determine the differential pressure between 280 and 400, 400 and 500 and 280 and 500 diameters downstream of the inlet, respectively.

A comparison between the DP cell and the gauge pressure differential measurements is depicted in Figure 3 for both air–water and air–oil mixtures. This comparison shows that the agreement is good and gets better as the pressure drop increases. For the DP cell, the accuracy is higher as based on specifications. It is a differential pressure transducer (Siemens Sitrans P DS III) with an operational range of 0 to 1000 Pa and an accuracy of 0.075% of the full range, whose pressure tapings are 3 m apart. Taking this as a reference, we can estimate that, in the worst-case scenario, the pressure drops from the gauge pressure transducers have an uncertainty within or better than 10% (in the lower range), which comes mainly as a result of the uncertainty propagation. In the world of multiphase flows, this level of uncertainty is not uncommon. In their review, Dukler et al. [3] found that, for apparently similar test conditions, pressure drop data from different investigators vary by 30 to 60%. Ortiz-Vidal et al. [34] reported pressure drop data with uncertainty within $\pm 8.5\%$ depending on the flow conditions. Hamad et al. [22] compared single-phase pressure drop data with the Blasius equation and determined that the error associated with the pressure drop measurements is within the range ($\pm 10\%$). This shows the importance of reliable experimental data in improving the performance of prediction methods.

The pressure in the flow loop was kept close to atmospheric by fully opening the control valve (CV6) on the separator (see Figure 1). At very high liquid and gas flow rates, the pressure reaches nearly 2 Bars due to the pressure drop downstream the test section. The temperature was in the range between 29 and 31 °C. Once the flow stabilized, data from all instruments were recorded simultaneously at a sampling frequency of 10 Hz over a time interval of 180 s. The measurements were supplemented using a high-speed video camera to visualize the flow in some runs. The fluid properties are listed in Table 1. The same oil has been previously used by Loh and Premanadhan [32]. This oil is considered as light oil due to its properties, and its viscosity was measured using an HAAKE MARS III Rheometer.

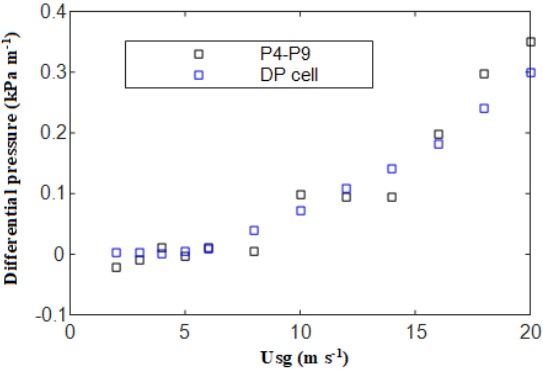

**Figure 3.** Comparison between pressure drop data from two gauge pressure sensors (P4−P9) and the DP cell.

**Table 1.** Physical fluid properties.

| Fluid | Density (kg m$^{-3}$) | Viscosity (kg m$^{-1}$ s$^{-1}$) | Surface Tension (N m$^{-1}$) |
|---|---|---|---|
| Air | 1.224 | 0.000018 | 0.072 |
| Tap water | 1000 | 0.001 | |
| AP process oil | 845 | 0.030 | 0.037 |

*2.2. Single Phase Measurements*

Since the pressure drop depends also on the inner surface roughness of the pipe material, $\varepsilon$, single-phase measurements were performed to verify the pipe friction coefficient, as displayed in Figure 4. A comparison with the standard value for smooth pipes shows a fairly good agreement when plotted against the Re number. Therefore, Blasius-type power-law expressions for the wall shear stresses can be used in the prediction models. These measurements show that the relative roughness, $\varepsilon/D$, is in the range of 1 to $2 \times 10^{-5}$. It can be seen that, according to the Blasius equation, the single-phase friction factor for oil is bigger than for water as it has a lower Reynolds number.

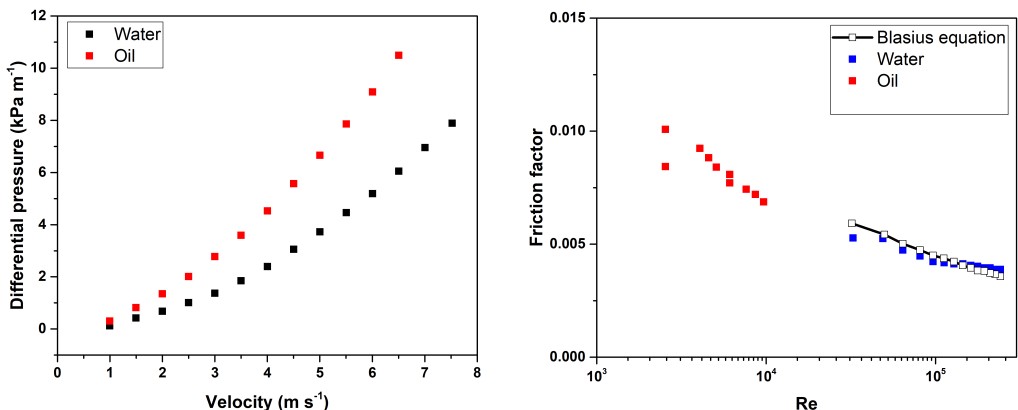

**Figure 4.** Single−phase pressure drop (**left**) and pipe friction coefficient (**right**) for oil and water.

## 3. Results and Discussion

A total of 254 data points were acquired and analyzed. The same flow conditions were set for both the air–water and air–oil mixtures. These flow conditions are similar to the ones commonly found in the oil and gas industry. In fact, similar flow rates in this flow loop have been used to test the performance of industrial multiphase flow meters for independent companies. They are also within the range where accelerational pressure drop can be neglected (see Ferguson and Spedding [5]). Thus, the measurements obtained correspond directly to the frictional pressure drop.

To investigate the pressure profiles along the pipeline, the measurements were made at axial locations of 280, 400 and 500 pipe diameters downstream of the inlet. The change in gauge pressure along the length of the test section is shown in Figure 5. It can be seen that the pressure gradient is fairly linear. Thus, the results obtained from P4–P7, P7–P9 and P4–P9 are consistent. Similar pressure profiles were obtained by Mattar and Gregory [23] for air–oil flow using differential pressure transducers whose low-pressure ports were joined together. Unfortunately, the gauge pressure P7 was not taken for all runs, as a DP cell was installed instead for low pressure drops at low superficial velocities using the same analog input channel in the data acquisition.

The automatic feature of the control system allows us to specify the gas and liquid superficial velocities directly during run time and based on the test section. However, the values of the gas superficial velocities are corrected with the pressure at the midpoint between the pressure taps by linearly interpolating the pressure from the P4 using the pressure gradient. Figure 6 shows a comparison of the pressure drops obtained by three

different combinations of subtraction, namely P4–P7, P7–P9 and P4–P9. It can be observed that the values are quite similar, which gives confidence in the results. The value of P4–P9 is taken as the ultimate result for further analysis. Since the pressure drop is no more than 5 kPa m$^{-1}$, the pressure change is less than 5%. In passing, we note that the pressure drop for the air–oil mixtures behaves differently from that for the air–water mixtures. This difference can be explained partially in terms of the relative viscosities for both liquid phases. The oil under consideration has a viscosity ten times higher than that of the tap water used in the experiments. From a dynamical point of view, all gas–liquid interactions take place at the interface, where the local stress state varies according to the flow conditions there. When the apparent velocities are large, this stress state may become sufficiently large to induce K-H instabilities. In this case, the intefacial flow enters what Andritsos et al. [36] refer to as the K-H subregime. It follows that the shearing effects lead to significant distortions in the interface, which ultimately change the properties of the flow itself (a coupled effect). This is due to the exacerbated exchange of momentum in all directions promoted by the thickening and undulation of the interface. The 'off-axis' exchange of momentum manifests itself in the overall pressure registered by the instruments installed along the pipe.

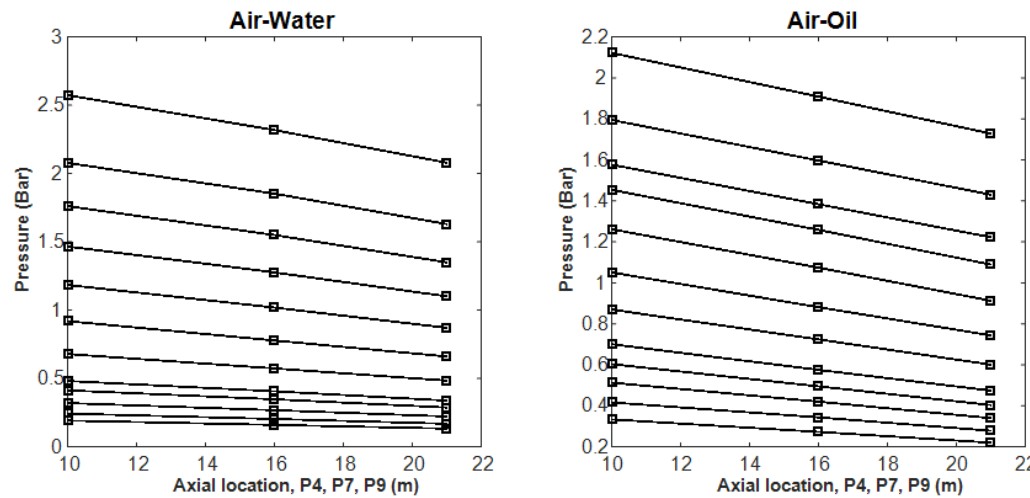

**Figure 5.** Example of typical pressure profiles for air–water (**left**) and air–oil (**right**) along the test section for different fixed flow conditions.

Figure 7 shows the pressure gradient as a function of the gas superficial velocity for varying values of the liquid superficial velocity. This allows other researchers to easily use these data and check if the pressure drop measurements are within the realistic range of values having consistent trends with neighbouring points. Compared to the air–water case, the air–oil flow exhibits a different pressure drop behavior, which is more evident in the intermediate velocity range between 0.25 and 1.5 m s$^{-1}$. It appears that the oil viscosity is such that a thicker layer of fluid remains adhered to the inner wall of the pipe after passage of a slug. At low liquid velocities, the interfacial stresses prevent the K-H instability from developing unsteadily, resulting only in a small undulating interface. However, as the liquid flow velocity increases, the interfacial stresses also increase, the effective cross-section is reduced as the layer becomes thicker and the K-H instability has more chances to develop. This results in a severe deformation of the thick oil layer around the wall, which, in turn, promotes flow separation and severe turbulence in the gaseous phase. These effects may induce adverse pressure gradients in the flow, resulting in an overall drastic reduction in the pressure drop along the pipe. In contrast, at high mixture velocities, the interfacial stresses are high enough to allow liquid removal from the layer. The removed fluid is dragged into the core of the flow, thereby leading to an effective reduction in the layer

thickness. This marks a kind of transition where the overall pressure gradient is restored to a steady average value.

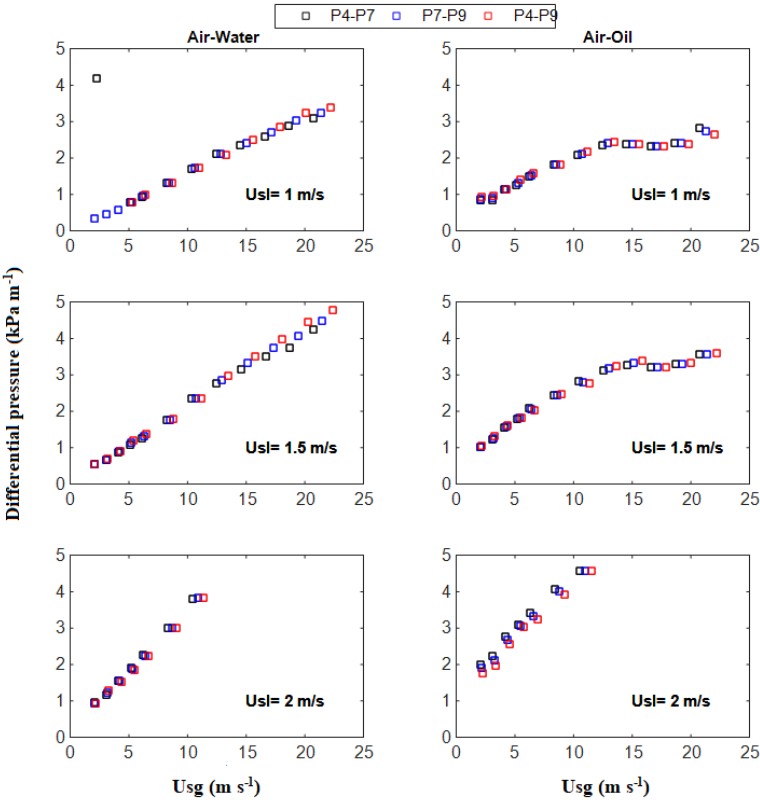

**Figure 6.** Comparison of pressure drop from P1−P7, P7−P9 and P4−P9 for air−water (**left**) and air−oil flows (**right**) in 2−inch line at different flow conditions of gas and liquid superficial velocities.

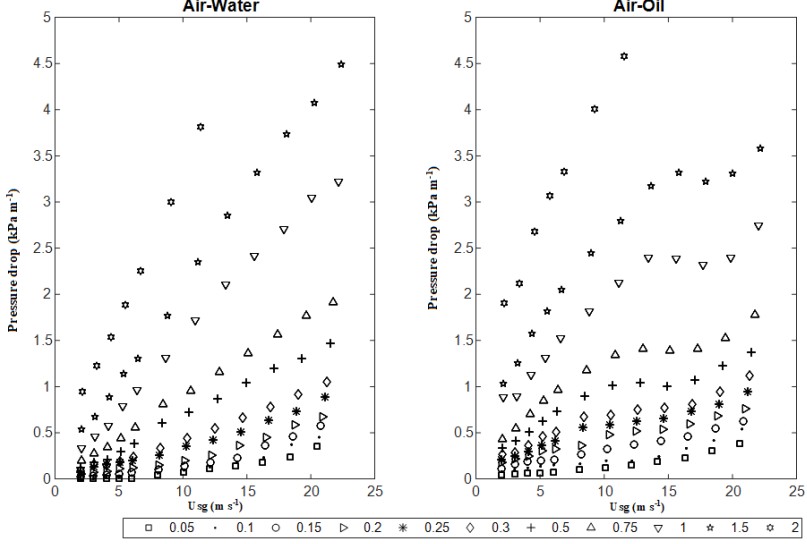

**Figure 7.** Pressure drop results for air−water (**left**) and air−oil flows (**right**) in 2−inch line at different flow conditions of gas and liquid superficial velocities.

In general, the effect of increasing the gas and liquid superficial velocities is to increase the pressure gradient. Also, for gas superficial velocities lower than around 10 m s$^{-1}$, as expected, the pressure drop is generally higher for the air–oil flow, since it is more viscous than water. Surface tension and density are less relevant due to the relatively big pipe diameter and horizontal pipe orientation. However, it can be noted that, for superficial

velocities beyond around 10 m s$^{-1}$, the pressure drop in air–oil flow can be lower than that in air–water flow. This is interesting because, as the viscosity increases, the pressure drop normally increases, as was reported by Gokcal et al. [25]. An effort was devoted to ensure the correctness of this result. Figure 8 shows a side-by-side comparison between the air–oil and air–water pressure drop, whereas Figure 9 shows a comparison with pressure drop data provided by Gokcal et al. [25], where the effects of viscosity can be appreciated. To explain this, we resort to flow visualization with a high-speed video camera and the use of a transparent section. In addition, other literature data were used for comparison. The experimental data reported in Figure 9 differ from those of Gokcal et al. [25] because both data sequences correspond to different viscosities. In the present work, the viscosity was 0.03 Pa s against 0.181 Pa s in the study by Gokcal et al. [25].

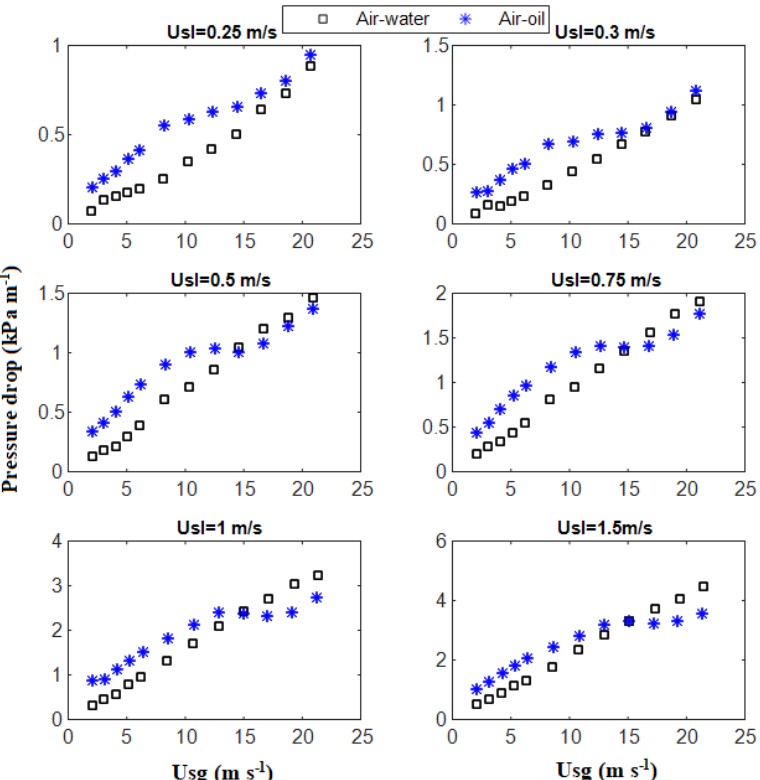

**Figure 8.** Comparison of pressure drop between air−water and air−oil flow for different liquid and gas superficial velocities.

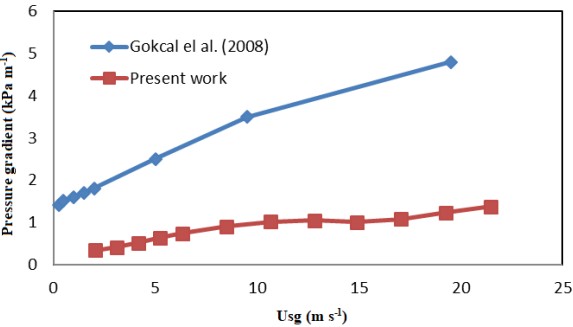

**Figure 9.** Comparison between Gokcal et al.'s [25] pressure drop measurements for air−oil flow with oil viscosity of 0.181 Pa s and present work. The data plotted correspond to a liquid superficial velocity of 0.5 m s$^{-1}$.

### 3.1. Flow Visualization

Figure 10 shows snapshots from the videos taken at a liquid superficial velocity of $0.5 \text{ m s}^{-1}$ with the high-speed camera at 250 fps. The full videos are available as complimentary material in the appendix. The videos show that these conditions correspond to intermittent flow. However, we focus on the section of the stratified liquid layer to explain the results of the pressure drop comparison. The picture for air–water (on the left column) shows that the interface appears more blurred due to the highly disturbed flow. For air–oil (on the right column), the interface is smooth with a greater height and the film around the pipe wall is visibly thicker. Similar observations are described by Andritsos and Hanratty [37] for a viscosity of 80 cp, although no clear pictures were provided. The wavier interface suggests that there is a bigger interfacial friction factor in the air–water interface. The presence of waves at the interface can cause the interfacial shear stress to be much greater than the one that would be observed if the interface were smooth [36]. The presence of fully developed roll waves was also found to increase interfacial friction at the gas–liquid interface [38]. The friction at the interface is often represented as a function of the wall friction of the gas, $fi = \varphi fg$. Andritsos et al. [36] proposed empirical correlations for the ratio $fi/fg$ for the 2D wave and Kelvin–Helmholtz (K-H) wave stratified sub-regimes. These regimes are characterized by significant distortions in the gas–liquid interface. Typically, these distortions correspond to wave flow patterns, which are promoted by the increased interfacial stresses caused by the increased slip velocity at the interface. According to Andritsos et al. [36], the interfacial friction factor can be up to 10x that of the gas phase in the K-H wave region. In the present experiments, the flow pattern reaches the annular flow region. As a result, larger pressure drops and lower liquid holdups are expected for the air–water case. Visual observations from the corresponding videos, with tiny bubbles trapped, reveal that the fluid in the stratified liquid layer seems to travel at a lower velocity for the case of oil compared to the air–water case. This explains the lower pressure drop obtained for the air–oil flow because the pressure drop depends on both the velocity and interfacial friction factor.

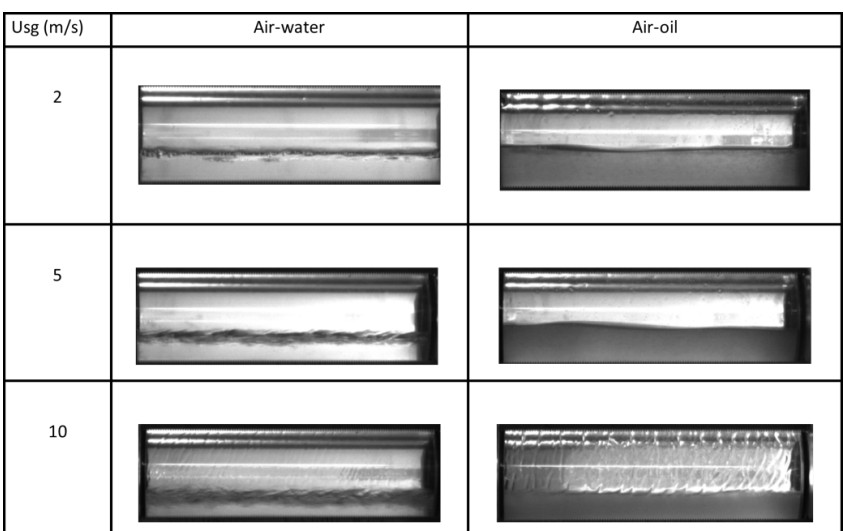

| Usg (m/s) | Air-water | Air-oil |
|:---:|:---:|:---:|
| 2 | | |
| 5 | | |
| 10 | | |

**Figure 10.** Visual comparison between the stratified liquid layers of air−water (**left column**) and air−oil flows (**right column**) for different gas superficial velocities of 2, 5 and 10 $\text{m s}^{-1}$. The liquid superficial velocity is $0.5 \text{ m s}^{-1}$.

On the other hand, Cohen and Hanratty [39] have proposed that the waves receive their energy from the gas through the work of pressure and shear stress perturbations induced by the waves. According to them, knowledge of the physical processes responsible for the generation of these waves is needed if one is to understand many of the phenomena that are observed in two-phase flow systems. However, most interfacial friction correlations consider knowledge of the liquid layer height instead of the fluid properties, which limits

their predicting capabilities. The air–oil region shows a greater viscous sub-layer thickness, implying that more regions of the flow are affected by viscous dissipation. The pressure disturbances caused by the shearing action of the gas over the liquid do not penetrate deeper into the fluid film for oil flow, due to the viscous dissipational effects. Hence, a smoother interface prevails in comparison to the disturbed interface in air–water flow.

The modeling of the interfacial momentum transfer is considered to be the crucial issue in gas–liquid stratified flows [40]. Andritsos and Hanratty [37] developed a model for the interfacial friction factor. A more rigorous analytical description is beyond the scope of this work and might be the subject of a separate future paper. It can be concluded from the visualization that the observed flow behavior is consistent with the pressure drop measurements. The visual observations suggest that, for low gas velocities, the oil wall friction overcomes the interfacial friction in the water. As the gas flow rate increases, the water interfacial friction eventually overcomes the wall friction of oil flow. This happens in the intermediate region between the annular and slug flows. Consequently, two different trends of the pressure drop versus the gas superficial velocity can be obtained for low and high gas superficial velocity. Footage was taken for different flow conditions for both the air–water and air–oil flows. The liquid stratified layer is consistently thicker for oil across the stratified and intermittent flow patterns. This suggests that, during intermittent flow, the flow characteristics, such as the slug translational velocity and frequency, might also be different. This information can be obtained from the recorded footage of the flow. To look into this, an image processing algorithm was used, similar to that of Loh et al. [32]. The algorithm tracks the gas–liquid interface from the footage to obtain the time trace of the liquid layer height. The time series was obtained at 250 Hz and then a low-pass filter was applied to eliminate the noise. Since this technique is tedious, only a handful of data points are indeed presented.

To calculate the frequency, a slug-counting algorithm was developed. The power spectral density (PSD) technique did not work very well because the slugs were nonuniform and/or not periodic. Therefore, several different values of the frequency would result from this technique. For the translational velocity, the cross-correlation velocity was applied, which works because the time traces of the liquid layer height are strongly correlated. The results are shown in Figures 11 and 12. It can be noted that the slug frequency is higher for the air–oil flow compared to the air–water flow, whereas the translational velocity is roughly the same for both flows. The time series of the average liquid holdup at a cross-sectional area of the pipe were obtained using image processing techniques. The side-view images were obtained with a time resolution of 0.004 s. The quality of the image will then determine the accuracy of the results. For example, high-quality images allow the algorithm to identify the gas–liquid interface clearly due to the color contrast. This has been attained using proper illumination on the transparent test section. Due to the complex nature of the flow patterns, it is not always possible to obtain very clear images and so we only apply the algorithm to flow conditions for which the interface can be identified.

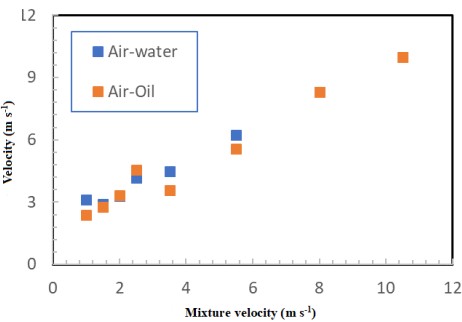

**Figure 11.** Comparison of the slug translational velocity between air−water and air−oil flows. The liquid superficial velocity is 0.5 m s$^{-1}$.

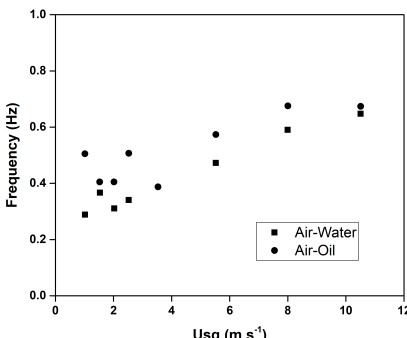

**Figure 12.** Comparison of the slug frequency between air−water and air−oil flows. The liquid superficial velocity is 0.5 m s$^{-1}$.

### *3.2. Comparison with Literature Data*

As was mentioned in the introduction, pressure drop data for air–oil two-phase flows are scarce. Nevertheless, owing to industrial demands, the research community has preferred to investigate three-phase flows. Although three-phase flows are beyond the scope of this work, some scenarios can be used to compare our results on two-phase flows. This is because the air–oil flow is a particular case of three-phase flow in which the water cut (fraction by volume of water in the liquid phase) is 0%, and it is 100% for air–water. A comparison of the present data against independent relevant data compiled from the open literature is displayed in Figures 13 and 14).

Hewitt [41] reported some results for three-phase flows obtained on the Imperial College WASP facility for high and low gas velocities (see upper two frames of Figure 13). These results include the pressure drop as a function of the input water cut. A keen look at his data for high gas velocities indicates that a 100% water cut would produce a higher pressure drop than a 0% water cut. However, for low gas velocities, this is not observed. The oil used in these experiments was a lubricating oil with a density of 860 kg m$^{-3}$ and a viscosity of around 40 cP (mPa s). The test section in this facility was a stainless steel pipe that was 38 m long and had a 77.92 mm internal diameter.

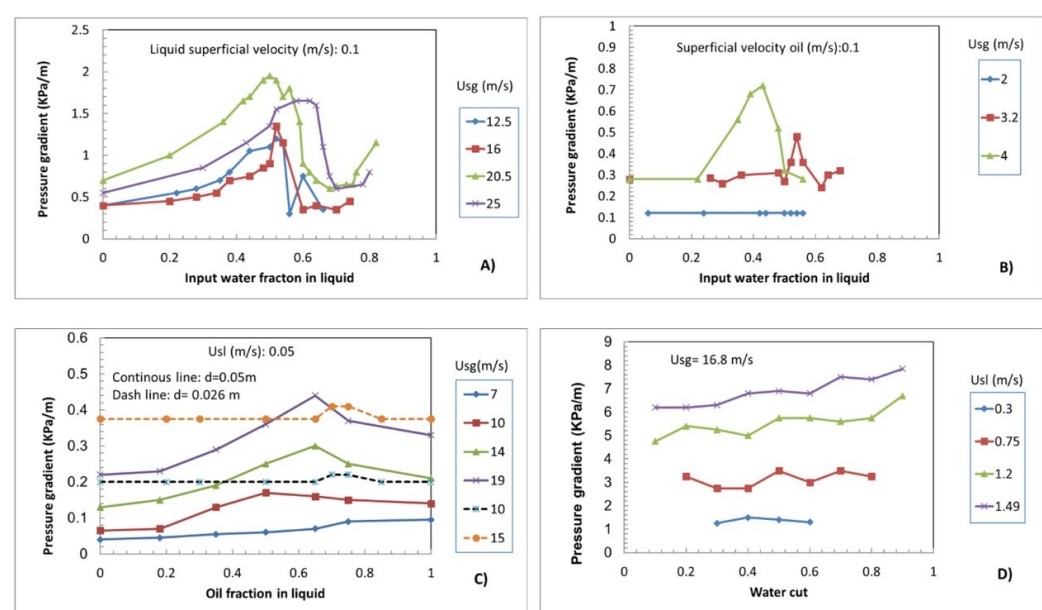

**Figure 13.** Experimental pressure drop data from three−phase flow experiments. The plots were reproduced from data reported by Hewitt [41] (**A**,**B**), by Spedding et al. [42] (**C**) and by Al-Hadhrami et al. [43] (**D**).

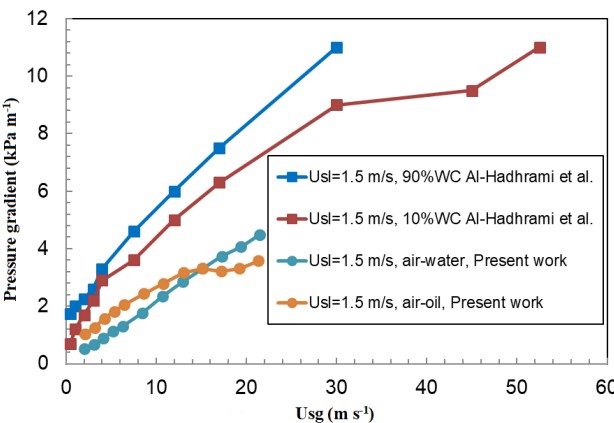

**Figure 14.** Pressure gradient as a function of the gas superficial velocity for the air−water and air−oil flow as compared with Al-Hadhrami et al.'s [43] experimental data for three−phase flow. In all cases, the liquid superficial veolocity is 1.5 m s$^{-1}$.

A further study was reported by Spedding et al. [42] (see bottom-left frame of Figure 13). The oil properties for the 0.0259 m ID facility were $\rho_0 = 828.5$ kg m$^{-3}$ and $\mu_0 = 0.0122$ kg m$^{-1}$ s$^{-1}$, while, for the 0.0501 m ID apparatus, they were $\rho_0 = 854.2$ kg m$^{-3}$ and $\mu_0 = 0.0395$ kg m$^{-1}$ s$^{-1}$, all at 24 °C. However, their results do not show the phenomenon observed in the present work, where the pressure drop for the air–water flow is higher than for the air–oil flow. In contrast, their liquid superficial velocities are limited to the low range compared to ours. In this range, we have also found a similar trend to theirs. In addition, what is interesting is that, for their 1-inch pipe diameter, the pressure drop is about the same for the mixture with a 0 and 100% water cut. Recently, Al-Hadhrami et al. [43] reported data for three-phase flow in a horizontal pipe at different water cuts, using Safrasol D80 oil with a density of 800 kg m$^{-3}$ and dynamic viscosity of 1.77 cp, tap water with a dynamic viscosity of 1 cp and density of 1000 kg m$^{-3}$ and air at standard conditions. Their section was an acrylic pipe of diameter equal to 22.5 mm. From the bottom-right frame of Figure 13, it can be seen that there is a clear trend of the pressure drop increasing with the water cut and the liquid superficial velocity.

Figure 14 compares the pressure gradient as a function of the gas superficial velocity of Al-Hadhrami et al.'s [43] data for a 10 and 90% water cut with present work for air–water and air–oil flow. Although Al-Hadhrami et al. [43] did not report data for pure air–oil and air–water flows, their data clearly show that the pressure drop is higher for a high water cut at high liquid superficial velocities in the whole range of gas superficial velocities. There is an agreement that the data for water show a higher pressure drop than the data for oil. It is worth noting that their oil viscosity is very low. However, their corresponding higher values can be attributed to their smaller pipe diameter. The difference with Al-Hadhrami et al.'s [43] data can be attributed to the fact that they used three-phase mixtures with different water cuts. Therefore, when the water content is bigger, the pressure is correspondingly higher. However, both data sequences predict an increase in the pressure gradient with the surface gas velocity, which indicates that our results are consistent.

From a comparison with experimental data available in the open literature, it can be concluded that there is implicitly enough independent evidence that the pressure drop for air–oil flows is lower than for air–water flows. However, from Gokcal et al.'s [25] data, it follows that, for high-viscosity oil, the lower pressure drop in air–oil flow might or might not happen at higher velocities. This apparently occurs because the wall friction in the high-viscosity oil is already too large to be overcome by the interfacial friction at the air–water interface. Furthermore, as shown in Figure 8, the pressure drop increases with increasing liquid superficial velocity because of the bigger interfacial area associated with a higher liquid stratified layer. This demonstrate the extreme complexity of multiphase flows and the need for a more basic understanding of them. This feature has important implications

for oil and gas pipelines with high throughput, where three-phase (i.e., gas–water–oil) mixtures are normally handled. For instance, separating the water from the gas and oil may result not only in an overall operating cost saving but also in facilitating the transport of the gas–oil mixture.

### 3.3. Comparison of Correlations

The performance of the pressure drop correlations for two-phase flows, such as those that were reported in the introduction, were compared against the experimental database plotted in Figure 4. There are many correlations available in the literature. However, a selection has been made here of the most widely used and accepted models. Nine different pressure drop calculation methods were selected. The methods selected were compared by determining the deviation between predicted and measured pressure drops. The models and correlations for the prediction of the pressure drop are summarized in Table 2 and the results of the comparison are presented in Figures 15–17.

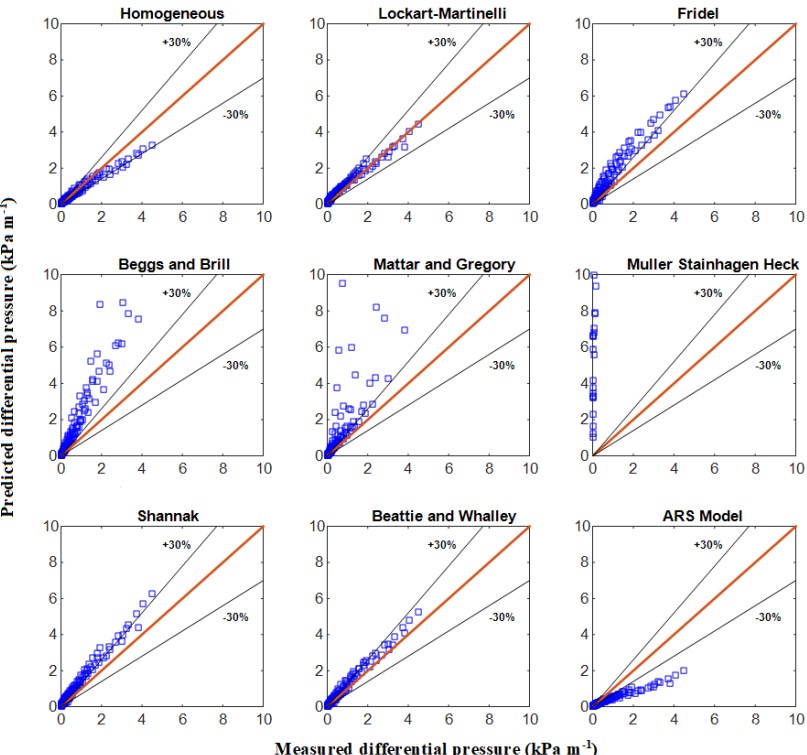

**Figure 15.** Comparison of the measured pressure drop data with the predictions of different correlations from the literature for air–water two–phase flow.

Owing to its simplicity, it is worth testing the homogeneous model given by Equation (1). One limitation in computing the two-phase frictional pressure gradient using the homogeneous modeling approach is the definition of the two-phase viscosity. It can be seen from Figure 15 that it underestimates the pressure drop for air-water, while, as shown in Figure 16, it overestimates the pressure drop for air-oil in the higher range. On the other hand, the Lockhart-Martinelli correlation, given by Equations (2)–(4), was developed based on experimental data in small-diameter pipes and including different two-phase mixtures consisting of air with benzene, kerosene, water and several oils. This correlation is limited by the viscosity, which will only affect the Reynolds number and consequently the value of the coefficient $C$. However, there are only two possible values of $C$ as a function of the liquid Re. It follows from Figure 15 that, for air-water mixtures, the predictions are around the right place, while, for air-oil mixtures, the Lockhart-Martinelli correlation overpredicts the experimental data by up to 100%.

**Table 2.** Selected frictional pressure drop models and correlations.

| Model Correlation | Equations |
|---|---|
| Homogeneous | Equation (1) |
| Lockhart-Martinelli (1949) [13] | Equations (2)–(4) |
| Friedel (1979) [14] | $\left(\frac{dp}{dL}\right)_{TP} = \phi_F^2 \left(\frac{dp}{dL}\right)_L$; $\phi_F^2 = E + \frac{3.24FH}{\text{Fr}_H^{0.045}\text{We}_L^{0.035}}$; $\text{Fr}_H = \frac{M^2}{gd\rho_H^2}$ $E = (1-x)^2 + x^2\frac{\rho_l f_g}{\rho_g f_l}$; $F = x^{0.78}(1-x)^{0.224}$ $H = \left(\frac{\rho_l}{\rho_g}\right)^{0.91}\left(\frac{\mu_g}{\mu_l}\right)^{0.19}\left(1 - \frac{\mu_g}{\mu_l}\right)^{0.7}$; $\text{We} = \frac{M^2 d}{\sigma\rho_H}$ |
| Beggs and Brill (1973) [12] | $\left(\frac{dP}{dL}\right)_f = \frac{2f_m\rho_{ns}U_m^2}{d}$; $\rho_{ns} = H_L\rho_L + (1-H_L)\rho_G$; $\mu_{ns} = \lambda\mu_L + (1-\lambda)\mu_G$; $f_m = f_{ns}e^s$; $\mathfrak{R}_{ns} = \frac{dU_m\rho_{ns}}{\mu_{ns}}$ $L1 = \exp\left(-4.62 - 3.757X - 0.481X^2 - 0.0207X^3\right)$ $L2 = \exp\left(1.061 - 4.602X - 1.609X^2 - 0.179X^3 + 0.635*10^{-3}X^5\right)$ $X = \ln\lambda$; $\lambda = \frac{U_{sl}}{U_{sl}+U_{sg}}$; $\text{Fr} = \frac{U_m^2}{gd}$ $\text{Fr} < L1\text{segregated}H_L = \frac{0.98\lambda^{0.4846}}{\text{Fr}^{0.0868}}$ $L1 \leq \text{Fr} \leq L2\text{Intermittent}H_L = \frac{0.98\lambda^{0.5351}}{\text{Fr}^{0.0173}}$ $L2 \leq \text{FrDistributed}H_L = \frac{0.98\lambda^{0.5824}}{\text{Fr}^{0.0609}}$ $f_{ns} = \left[2\log\left(\frac{\mathfrak{R}_{ns}}{4.5223\log\mathfrak{R}_{ns}-3.8215}\right)\right]^{-2}$ $s = \frac{\ln y}{-0.0523+3.182\ln y-0.8725(\ln y)^2+0.01853(\ln y)^4}$; $y = \frac{\lambda}{H_L^2}$; $s = (\ln 2.2y - 1.2)$ if $1 < y < 1.2$ |
| Mattar and Gregory (1974) [23] | $\left(\frac{dP}{dL}\right)_f = \frac{\rho_m g\sin\theta + (2f\rho_{mE}U_m^2)/d}{1-(\rho_m U_m U_{sg})/P}$; $\rho_{mE} = H_L\rho_L + (1-H_L)\rho_G$ $f = 0.0014 + \frac{0.125}{(\mathfrak{R}_{mE})^{0.32}}$; $E_L = 1 - \frac{U_{gs}}{1.3(U_{sg}+U_{sL})+0.7}$ |
| Müller-Steinhagen and Heck (1986) [9] | $\left(\frac{dP}{dL}\right)_f = G(1-x)^{1/3} + Bx^3$; $G = A + 2(B-A)x$ $A = \left(\frac{dP}{dL}\right)_{fl} = \frac{f_l M^2}{2\rho_l d}$ $B = \left(\frac{dP}{dL}\right)_{fg} = \frac{f_g M^2}{2\rho_g d}$ |
| Shannak (2008) [44] | $\left(\frac{dP}{dL}\right)_f = \frac{f_{TP}M^2}{2\rho_{tp}d}$ $\frac{1}{\sqrt{f_{tp}}} = -2\log\left[\frac{1}{3.7065}\frac{\epsilon}{d} - \frac{4.0452}{\mathfrak{R}_{tp}}\log\left(\frac{1}{2.8257}\left(\frac{\epsilon}{d}\right)^{1.1098} + \frac{5.8506}{(\mathfrak{R}_{tp})^{0.8981}}\right)\right]$ $\mathfrak{R}_{tp} = \frac{F_{lg}+F_{lf}}{F_{vg}+F_{vl}}$; $F_{lg} = \rho_g v_g^2 d^2$; $F_{lf} = \rho_f v_f^2 d^2$; $F_{vg} = \rho_g v_g^2 d^2$; $F_{vf} = \rho_f v_f^2 d^2$ |
| Beattie and Whalley (1982) [45] | $\left(\frac{dP}{dL}\right)_f = \frac{f_{TP}M^2}{2\rho_{tp}d}$; $\mathfrak{R} = \frac{Gd}{\mu}$; $\mu = \mu_l(1-\beta)(1+2.5\beta) + \mu_g\beta$; $\beta = \frac{\rho_l x}{\rho_l x + \rho_g(1-x)}$ |
| Apparent rough surface (ARS) model, Hart et al. (1989) [17] | $\frac{H_L}{1-H_L} = \frac{Usl}{Usg}\left\{1 + \left[10.4 + \mathfrak{R}_{sl}^{-0.363}\left(\frac{\rho_l}{\rho_g}\right)^{1/2}\right]\right\}$; $u_l = \frac{Usl}{H_L}$; $u_g = \frac{Usg}{1-H_L}$; $\text{Fr} = \frac{u_l^2\rho_l}{gD\Delta\rho}$; $\mathfrak{R}_g = \frac{du_g\rho_g}{\mu_g}$ $\theta = \theta_0 + 0.26\text{Fr}^{0.58}$; $\theta_0 = 0.52 + 0.26H_L^{0.374}$; $\frac{k}{d} = 2.3\frac{H_L}{4\theta}$ $f_i = \frac{0.0625}{\left[\log_{10}\left(\frac{15}{\mathfrak{R}_g} + \frac{k_g}{3.715d}\right)\right]^2}$; $f_g = \frac{0.07725}{\left[\log_{10}\left(\frac{\mathfrak{R}_g}{7}\right)\right]^2}$; $f_{TP} = (1-\theta)f_g + \theta f_i$ $\left(\frac{dP}{dL}\right)_{TP} = \frac{1}{1-H_L}\left[4f_{TP}\left(\frac{1}{2d}\right)\rho_g u_g^2 - 4\theta f_i\left(\frac{1}{2d}\right)\rho_g(2u_g u_l - u_l^2)\right]$ |

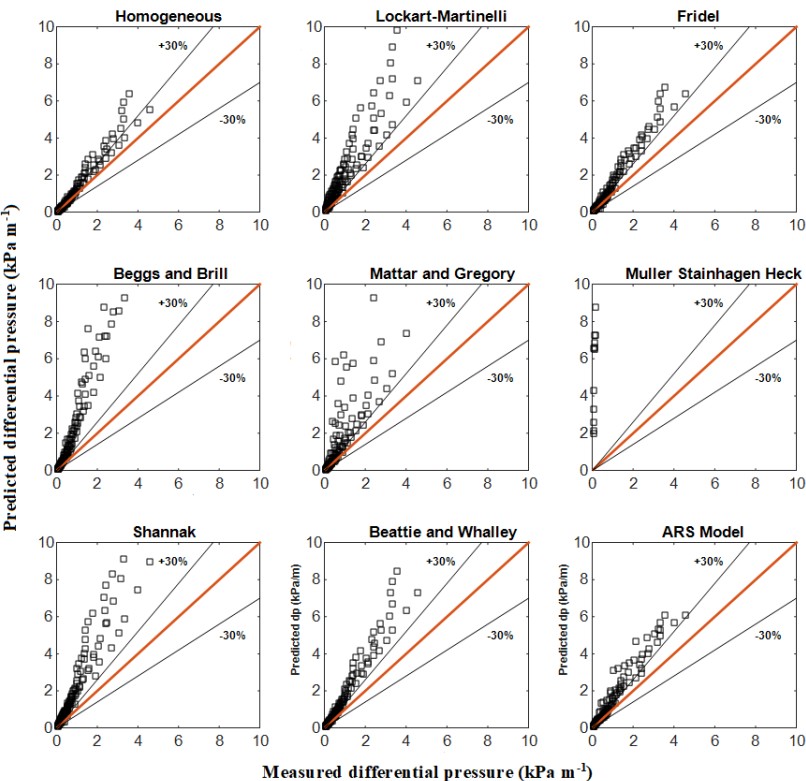

**Figure 16.** Comparison of the measured pressure drop data with the predictions of different correlations from the literature for air−oil two−phase flow.

An alternative method of calculating the two-phase flow multiplier is employed in the Friedel [14] correlation, which is based on one of the biggest data banks in the literature (16,000 data points). These include a wide range of two-phase systems (single and two components), mass fluxes, mass flow rates, density ratios, viscosity ratios, surface tension and hydraulic diameters of circular, rectangular and annular tubes, as well as horizontal flows, vertical up and down flows and, in particular, exhaustive air–oil data. With both the Chisholm [8] and Friedel [14] multipliers, the separated flow model appears to give reasonable results, particularly in the low pressure drop range, where the superficial velocities also happen to be the lowest. The reason for this could be attributed to the fact that, in this situation, the models seem to be more sensitive to small changes in some of the parameters involved in the overall calculation. For instance, in this case, the Lockhart–Martinelli parameter becomes very small and the two-phase multiplier increases, which gives as a result a higher predicted pressure drop. The way that the friction factor is calculated can also have an effect.

The Beggs and Brill correlation [12] was developed for inclined pipes based on air–water mixtures. Zhao et al. [27] found good agreement using pipes of 26 mm diameter at low viscosity. However, as can be observed from Figures 15 and 16, the Beggs and Brill correlation fails by quite a bit and is also too complicated. It overpredicts the pressure drop by about 100–300% even for air–water flows. However, it predicts reasonably well the pressure drop for inclined flow with high liquid holdups, especially in pipes of 38 mm diameter. Moreover, the correlation developed by Mattar and Gregory [23] seems to work reasonably well in the range of superficial velocities below 10 m s$^{-1}$. Beyond that limit, it yields randomly scattered values differing by several orders of magnitude (not shown here). This can be attributed to the fact that this correlation was developed based on the concept of the drift flux model for intermittent flow (slug and plug flows) and tested by the authors using only air-flow data.

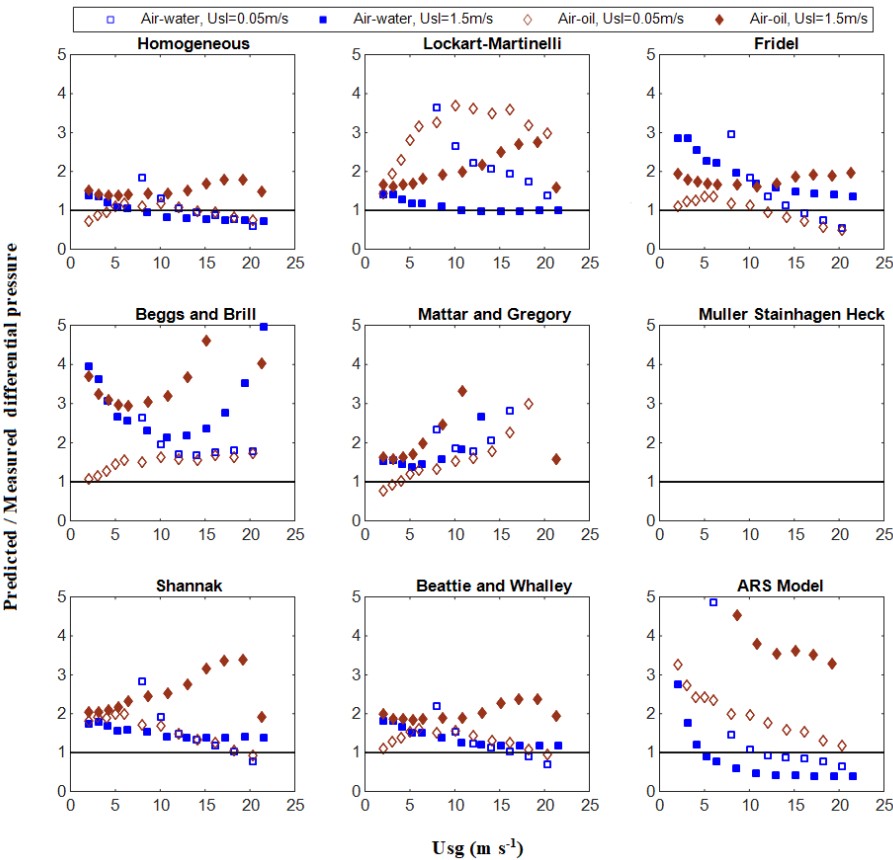

**Figure 17.** Effect of the gas superficial velocity on the ratio of predicted to experimentally measured pressure drops for different correlations.

As was stated by the authors, the correlation of Muller-Steinhagen and Heck [9] can be used only as long as the frictional pressure drop in the gas flow is higher than the frictional pressure drop in the liquid flow. Therefore, it is not applicable in the range spanning the present data. It was included here for the sake of completeness. It has been highly recommended by Xu et al. [19] and other authors, whose database was limited to pipe diameters of less than 14 mm and no oil data were included. As shown in Figures 15 and 16, this correlation fails to predict the correct pressure drops by several orders of magnitude. In passing, we note that one difference between small and large diameters might be the accelerational contribution to the pressure drop. A further correlation that is worth testing is the Shannak [44] correlation, which, in addition to being based on a simple concept, has been tested favorably against Friedel's database. According to Figures 15 and 16, this correlation overestimates the pressure drop in air–water flows by 30% and in air–oil flows by about 50%. In a particular way, it follows a similar trend to that of Friedel's [14] correlation for the case of air–water flow. The large errors exhibited by the different correlation models are expected because the experimental setup used in this work and the experimental conditions employed by the different authors are not the same. Even slight variations in the setup conditions may alter the nature of the flow since flow pattern transitions may occur at different operating points. For instance, the errors for the air–oil flows are far more significant than those for the air–water flows, while most correlations are approximately valid in the limit of low viscosities. This implies that viscosity plays a fundamental role in bringing about such discrepancies and that perhaps the inclusion of a correction factor to account for higher viscosities may help in reducing the errors. However, we do not have enough information at hand to recommend any improvements of this kind.

A further simple correlation was proposed by Beattie and Whalley [45] in which the flow pattern dependence is allowed in an implicit manner. This correlation is of a

homogeneous type that uses a hybrid model to define the viscosity. Also, it uses the same equation for all values of the Reynolds number. From an inspection of Figures 15 and 16, the overall performance of this model is only second to the standard homogeneous model. Finally, the apparent rough surface (ARS) model developed by Hart et al. [17] was designed for very low liquid holdups. This model performs predictions of the pressure drop in horizontal pipe flows by using a modified expression for the pressure drop in the gas phase. As expected, it provides fairly good predictions only at flow conditions of low liquid and high gas superficial velocities. A quantitative assessment of the correlations plotted in Figures 15 and 16 is given in Table 3 in terms of the average relative error (ARE)

$$\mathfrak{R} = \frac{1}{n} \sum^{n} \left| \frac{(dp/dL)_{\text{pred}} - (dp/dL)_{\text{exp}}}{(dp/dL)_{\text{exp}}} \right| \times 100. \tag{5}$$

The prediction errors within 20 and 50% are also included in Table 3. In this case, the higher the value, the better the prediction performance. It follows that the homogeneous model gives the best overall results. However, if a particular scenario is considered, the result is slightly different. For air–water flows, the predictions of the Lockhart–Martinelli correlation is the best, while, for air–oil flows, the homogeneous correlation performs better. For low liquid at high gas superficial velocities, the ARS model is better. In general, for air–oil flows, the correlations tested perform worse than for air–water flows.

**Table 3.** Performance of pressure drop correlations.

| Model | Overall | | | Air–Water ARE (%) | Air–Oil ARE (%) |
|---|---|---|---|---|---|
| | ARE (%) | ARE (%) <20% | ARE (%) <50% | | |
| Homogeneous | 54.59 | 47.43 | 84.98 | 84.79 | 24.14 |
| Lockhart–Martinelli (1949) [13] | 143.19 | 12.25 | 36.36 | 163.99 | 122.22 |
| Friedel (1979) [14] | 149.43 | 10.67 | 45.45 | 254.38 | 43.64 |
| Beggs and Brill (1973) [12] | 180.63 | 0.79 | 7.51 | 219.56 | 141.39 |
| Mattar and Gregory (1974) [23] | 607.88 | 13.83 | 35.57 | 314.93 | 903.16 |
| Muller-Steinhagen and Heck (1986) [9] | 16,891.84 | 0.00 | 0.00 | 17,872.55 | 15,903.34 |
| Shannak (2008) [44] | 157.42 | 3.16 | 18.58 | 193.21 | 121.34 |
| Beattie and Whalley (1982) [45] | 91.92 | 15.42 | 45.85 | 120.43 | 63.18 |
| Apparent rough surface ARS model Hart et al. (1989) [17] | 228.44 | 11.07 | 33.20 | 82.19 | 375.86 |

As was pointed out by Dukler et al. [3], the standard deviation is a completely descriptive measure of the spread between predicted and measured values only if the population of points follows a normal distribution. Furthermore, the results are biased if the data are not equally distributed. Therefore, the correlation performance is dependent on the experimental database. For instance, we can see that, for the case of air–water flow, the ARS model has the second best performance, which is due to the fact that more points were taken at low liquid and high gas superficial velocities.

On the other hand, if the predicted pressure drop is plotted against the experimental data, the errors are seen to grow at a higher pressure drop. However, it is not easy to see under which flow conditions the discrepancies happen. Figure 17 shows the ratio of predicted to experimental pressure drops as a function of the gas superficial velocity for both air–water and air–oil flows and all correlations listed in Tables 2 and 3. In general, for air–oil flows, the discrepancy between the predicted and experimental data grows as the gas superficial velocity increases. Note that only values of the ratio of predicted to experimental pressure drops up to 5 have been plotted in Figure 17. It is interesting to see that the correlation developed by Lockhart–Martinelli has a better performance than newer correlations.

Tribbe and Müller-Steinhagen [9] have shown that the precision and accuracy of phenomenological models are almost the same as those of empirical methods. However, because of their nature, the physically based models have more potential in the long term. Usually, when a new correlation or model is proposed, it is usually claimed to be superior to others. However, the present work, along with other independent test analyses, appears to not confirm this. In general, the experimental data also carry uncertainties, which are then transmitted to the proposed correlation. The degree of uncertainty may also depend on the measurement setup. In addition, since most correlations require information on the void fraction or flow pattern as input, errors in the void fraction accuracy and/or in the identification of the correct flow pattern are carried forward to the pressure drop prediction.

## 4. Conclusions

A comparative study of pressure drop in horizontal pipe flows has been carried out. A set of experimental data for air–water and air–oil mixtures has been reported. Based on these data, an evaluation of different models and correlations available in the open literature was performed. Flow visualization was also invoked to understand the results. The following conclusions can be drawn:

- The pressure drop increases as a function of the liquid and gas superficial velocities. This is particularly true for gas superficial velocities higher than $2 \text{ m s}^{-1}$ because, in these cases, the pressure drop is dominated by the frictional component.
- The pressure drop in air–oil flows can be lower than that in air–water flows. Using flow visualization to explain this phenomenon, it was observed that this is due to the higher interfacial friction that occurs in the case of air–water around the annular flow transition. This also results in a greater height of the stratified liquid layer for the oil.
- Independent data from the open literature implicitly agree with the above findings.
- The phenomenon observed has important implications for flow assurance in oil and gas pipelines with high throughput, where three-phase (i.e., gas–water–oil) mixtures are normally handled. This is true because separating the water may result either in an overall operating cost saving or even facilitate the transport of the gas–oil mixture.
- Within the experimental range considered in this work, it is interesting to see that the Lockhart–Martinelli [13] correlation performed better in general than newer correlations. However, the homogeneous model performed better than all other more complex correlations.
- In general, the correlations tested were seen to perform worse for air–oil flows than for air–water flows.

**Author Contributions:** Conceptualization, E.G. and V.H.P.; methodology, E.G. and V.H.P.; software, F.A.R.; formal analysis, V.H.P.; investigation, F.A.R.; resources, E.G.; data curation, L.S.; writing—original draft preparation, V.H.P.; writing—review and editing, L.S.; visualization, F.A.R.; supervision, J.K.; project administration, E.G.; funding acquisition, J.K. All authors have read and agreed to the published version of the manuscript.

**Funding:** This research was funded by the European Union's Horizon 2020 Programme under the ENERXICO Project under grant number 828947 and by the Mexican CONAHCYT-SENER-Hidrocarburos Programme under grant number B-S-69926.

**Data Availability Statement:** The data presented in this study are available on request from the corresponding author.

**Acknowledgments:** We acknowledge support given by the Laboratorio de Flujos Multifásicos IIUNAM.

**Conflicts of Interest:** The authors declare no conflict of interest.

**Abbreviations**

Nomenclature

| | |
|---|---|
| $A$ | Area, m$^2$ |
| $C$ | Chisholm coefficient |
| $d$ | Diameter, m |
| $dP/dL$ | Pressure drop per unit of length, Pa/m |
| $E$ | In situ phase volume fraction |
| $f$ | Friction factor |
| Fr | Froud number |
| $g$ | Gravity acceleration, m/s$^2$ |
| $G$ | Mass flux, kgm$^{-2}$s$^{-1}$ |
| $HL$ | Liquid holdup |
| $k$ | Average roughness of the tube wall |
| $L1$ | Correlation boundary |
| $L2$ | Correlation boundary |
| $M$ | Mass flow rate, kg/s |
| Re | Reynolds number |
| $u$ | Average phase velocity, m/s |
| $Um$ | Mixture velocity, m/s |
| $Usg$ | Superficial velocity of gas, m/s |
| $Usl$ | Superficial velocity of liquid, m/s |
| We | Weber number |
| $x$ | Quality |
| Subscripts | |
| $f$ | Friction |
| $g$ | Gas |
| $i$ | Interface |
| $l$ | Liquid |
| $m$ | Mixture |
| $ns$ | No-slip |
| $T$ | Total |
| $TP$ | Two-phase |

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
