# Peer review of "Comparative Study of Air–Water and Air–Oil Frictional Pressure Drops in Horizontal Pipe Flow"

_fluids, doi:10.3390/fluids9030067_

Round 1

Reviewer 1 Report

Comments and Suggestions for Authors

This study conducted experimental comparisons on the pressure drop of air water and air oil flow in horizontal pipes and evaluated the computational accuracy of existing models. This work is very meaningful. Agree to publish after making modifications to the following questions.

1. As shown in Fig. 6, when the apparent flow velocity is large, the flow pressure drop of air-oil shows a significant decrease, which is less explained in the manuscript and more details are suggested.

2. The sudden change in pressure drop of air oil flow in Fig. 7 varies under different experimental conditions (from 0.05 to 2). What is the reason for this? At the same time, it is recommended to provide a more detailed explanation on why the pressure drop increases after a significant decrease.

3. The experimental results of this study in Fig. 9 and Fig. 14 are quite different from those of Gokcal et al. and Al-Hadhrami et al. What are the main reasons?

4. As shown in Fig. 15 and Fig. 16, the calculation results of the existing models have  large errors. Can the main reasons for the large error in these models be further explained in the manuscript?

5. In the manuscript, the accuracy of the existing model is evaluated mainly through experimental data, and a lot of experimental data is also presented in the manuscript. Is it possible to use the large amount of experimental data in this study to modify the model to further improve the calculation accuracy of the flow pressure drop of air-water and air-oil in horizontal tubes?

Comments on the Quality of English Language

The English expression in the manuscript is appropriate.

Author Response

Page 8, lines 265 to 278, Point 1 of Referee 1

Page 8, lines 282 to 294, Point 2 of Referee 1

Page 9, lines 295 to 297, Point 2 of Referee 1

Page 9, lines 310 to 313, Point 3 of Referee  1

Page 14, lines 429 to 433, Point 3 of Referee 1

Page 18, lines 509 to 519, Point 4 of Referee 1

Reviewer 2 Report

Comments and Suggestions for Authors

please see the attached report

Author Response

Page 2, Equation 1. Point 1 of Referee 2

Page 7, Figure 4 (a minor point of Referee 2)

Page 10, lines 329 to 335, Point 2 of Referee 2

Page 12, lines 381 to 389, Point 3 of Referee 2

Page 14, Fig. 13 enlarged, Point 4 of Referee 2

Round 2

Reviewer 2 Report

Comments and Suggestions for Authors

The reviewer is fine with the updated version of the manuscript